# Roles of ASYMMETRIC LEAVES2 (AS2) and Nucleolar Proteins in the Adaxial–Abaxial Polarity Specification at the Perinucleolar Region in Arabidopsis

**DOI:** 10.3390/ijms21197314

**Published:** 2020-10-03

**Authors:** Hidekazu Iwakawa, Hiro Takahashi, Yasunori Machida, Chiyoko Machida

**Affiliations:** 1Graduate School of Bioscience and Biotechnology, Chubu University, 1200, Matsumoto-cho, Kasugai, Aichi 487-8501, Japan; iwakawa@isc.chubu.ac.jp; 2Graduate School of Medical Sciences, Kanazawa University, Kakuma-machi, Kanazawa, Ishikawa 920-1192, Japan; takahasi@p.kanazawa-u.ac.jp; 3Division of Biological Science, Graduate School of Science, Nagoya University, Furo-cho, Chikusa-ku, Nagoya, Aichi 464-8602, Japan

**Keywords:** *ASYMMETRIC LEAVES2*, AS2/LOB domain, adaxial–abaxial polarity, *ETTIN/AUXIN RESPONSE FACTOR3* (*ETT/ARF3*), AS2 body, nucleolus, gene body methylation, ribosomal DNA (rDNA)

## Abstract

Leaves of Arabidopsis develop from a shoot apical meristem grow along three (proximal–distal, adaxial–abaxial, and medial–lateral) axes and form a flat symmetric architecture. *ASYMMETRIC LEAVES2* (*AS2*), a key regulator for leaf adaxial–abaxial partitioning, encodes a plant-specific nuclear protein and directly represses the abaxial-determining gene *ETTIN/AUXIN RESPONSE FACTOR3* (*ETT/ARF3*). How AS2 could act as a critical regulator, however, has yet to be demonstrated, although it might play an epigenetic role. Here, we summarize the current understandings of the genetic, molecular, and cellular functions of AS2. A characteristic genetic feature of *AS2* is the presence of a number of (about 60) modifier genes, mutations of which enhance the leaf abnormalities of *as2*. Although genes for proteins that are involved in diverse cellular processes are known as modifiers, it has recently become clear that many modifier proteins, such as NUCLEOLIN1 (NUC1) and RNA HELICASE10 (RH10), are localized in the nucleolus. Some modifiers including ribosomal proteins are also members of the small subunit processome (SSUP). In addition, AS2 forms perinucleolar bodies partially colocalizing with chromocenters that include the condensed inactive 45S ribosomal RNA genes. AS2 participates in maintaining CpG methylation in specific exons of *ETT/ARF3*. *NUC1* and *RH10* genes are also involved in maintaining the CpG methylation levels and repressing *ETT/ARF3* transcript levels. AS2 and nucleolus-localizing modifiers might cooperatively repress *ETT/ARF3* to develop symmetric flat leaves. These results raise the possibility of a nucleolus-related epigenetic repression system operating for developmental genes unique to plants and predict that AS2 could be a molecule with novel functions that cannot be explained by the conventional concept of transcription factors.

## 1. Leaf Developments in Arabidopsis

Leaves develop from a shoot apical meristem (SAM) as lateral organs along three axes: proximal–distal, adaxial–abaxial, and medial–lateral [1,2,3,4,5,6,7]. Initially, groups of cells on the peripheral zone of the SAM are specified in leaf primordia (P0, Figure 1) and grow along the proximal–distal axis (P1). Then, adaxial–abaxial structures are differentiated (P2). Subsequently, cells proliferate along the medial–lateral axis leading to flat and symmetric leaves (Figure 1) [2,3,8]. To date, numerous genes involved in adaxial–abaxial determination have been reported in *Arabidopsis thaliana* [2,9]. The *ASYMMETRIC LEAVES2* (*AS2*) and *ASYMMETRIC LEAVES1* (*AS1*), which encode a protein with the plant-specific AS2/LOB domain and a protein with the MYB (SANT) domain, respectively, were originally identified as factors involved in symmetric leaf lamina formation [10,11,12,13]. Recent studies have revealed, however, that AS2 and AS1 regulate proper morphology along all three axes of leaves. The *Rough Sheath2* (*RS2*) gene of maize, an ortholog of *PHANTASTICA* (*PHAN*) of *Antirrhinum majus* and *AS1* of Arabidopsis, is involved in the proximal–distal patterning of maize leaves through the repression of class 1 *KNOX* genes [10,14,15]. The *PHAN* gene is involved in growth and the adaxial–abaxial determination of lateral organs. In addition, its activity is required early in the growth of leaves in the direction of the proximal–distal axis [16,17]. The ectopic expression of class 1 *Knotted1-like homeobox* (*KNOX*) genes in *as1* and *as2* mutant plants results in reductions in the growth of leaf blades and petioles in Arabidopsis, and these phenotypes are suppressed by mutations of the class 1 *KNOX* genes, *brevipedicellus* (*bp*), *knat2,* and *knat6*. These results indicate that the *AS1* and *AS2* genes of Arabidopsis are involved in the establishment of the proximal–distal axis through the repression of the class 1 *KNOX* genes [18]. In addition, the formation of shorter petioles and leaf blades in *as1* and *as2* is due to repression of gibberellin-synthetic genes by the upregulation of *BP*/*KNAT1*, *KNAT2*, and *KNAT6* [18]. AS1, acting together with AS2, directly represses the expression of the *BP* and *KNAT2* genes [19]. In this review, we focus on the establishment of leaf adaxial–abaxial polarity.

The *PHABULOSA* (*PHB*), *PHAVOLUTA* (*PHV*), and *REVOLUTA* (*REV*) genes encode class III homeodomain-leucine zipper (HD-ZIPIII) proteins, which determine adaxial cell fate [20,21,22]. Small RNAs play critical roles in specifying adaxial–abaxial polarity [23,24]. Micro RNAs miR165/166 promote the degradation of *HD-ZIPIII* transcripts in the abaxial domain, which results in the accumulation of HD-ZIPIII in the adaxial domain [24]. Members of the *KANADI* (*KAN*) gene family, which encode proteins with the GARP domain, determine abaxial cell fate [22,25]. The Arabidopsis genome contains six *YABBY* genes, which encode transcription factors with a zinc-finger domain and an HMG-related domain with a helix–loop–helix structure. The three *YABBY* genes, *FILAMENTOUS FLOWER* (*FIL*), *YABBY3* (*YAB3*), and *YAB2* are expressed in the abaxial domains of all leaf-derived organs, including cotyledons, leaves, and floral organs [26,27,28,29,30]. Furthermore, genetic analyses have shown that four *YABBY* genes (*FIL*, *YAB3*, *YAB2*, *YAB5*) govern embryo patterning and the growth of leaf lamina along the abaxial–adaxial boundary [30].

*ETTIN/AUXIN RESPONSE FACTOR3* (*ETT/ARF3*) and *ARF4* also specify both abaxial cell fate and the lateral growth of leaf lamina [31]. Transcripts of both *ETT/ARF3* and *ARF4* are specifically degraded by the small RNA tasiR-ARF, which is generated through a miR390 pathway in the presumptive adaxial domain and contributes to the determination of the adaxial cell fate [23]. Because a loss of adaxial–abaxial polarity is often accompanied by a defect of leaf lamina expansion, it is suggested that the lateral growth of the lamina could be related to the determination of adaxial–abaxial identity, as previously proposed [2,32].

## 2. Roles of AS2–AS1 in the Development of Leaf Polarity

As described above, AS2 and AS1 proteins, which have AS2/LOB and R2R3 MYB (SANT) domains, respectively (Figure 2a), are identified by a yeast two-hybrid system, pull-down and gel-shift assays, and subcellular co-localization analyses. Because of their nature, these experiments indicate that AS2 and AS1 are physically associated with each other in vitro [19,33,34,35,36], implying also that they form a protein complex in the nucleus. Transcripts of *AS2* and *AS1* genes accumulate throughout the entire leaf primordia at early stages, in which the AS2–AS1 complex might be formed, but the accumulation patterns change as the leaves develop [37]. *AS2* transcripts are detected in the adaxial domain, while *AS1* transcripts are detected in the central region between the adaxial and abaxial domains of leaf primordia and the vasculature regions in more developed leaf primordia [10,37]. The plant-specific AS2/LOB domain includes a CXXC-type zinc-finger (ZF) motif, a leucine-zipper-like (LZL) region, and the internal-conserved-glycine (IcG) region between ZF and LZL (Figure 2a). The AS2/LOB domain is highly conserved in the AS2/LOB family, which consists of 42 members including AS2 in Arabidopsis [12,38,39]. Since the amino acid sequences outside of the domains are diverged among members and the transcription patterns of these genes differ for each gene, the roles of these genes in Arabidopsis development seem to be distinct. Members that might retain functions similar to those of the *AS2* gene do not appear to exist in this family, because the substitution of the AS2/LOB domain of AS2 with those of other members disrupts its function [39]. Considering similarities among the AS2/LOB domains, it is, however, undeniable that these family members may retain partially overlapping functions at the molecular level. They have often been described as transcription factors [40,41,42,43,44,45,46,47]. Recent results on AS2, however, suggest that the term “transcription factor” is not appropriate for a member of this family; they are better described as novel functional factors that could play a role in gene expression.

Transcriptome analyses of *as2* and *as1* mutants reveal that accumulations of *ETT/ARF3*, *KAN2*, and *YAB5* transcripts, all of which are related to the abaxial cell fate, are increased in *as2* and *as1*, whereas those of the adaxial domain-determining *HD-ZIPIII* are not changed [37,48]. A subsequent systematic analysis has revealed that *ETT/ARF3* is a direct target of the AS2–AS1 complex [49,50]. AS2–AS1 directly represses *ETT/ARF3* by binding to the upstream region of *ETT/ARF3*. Furthermore, AS2–AS1 indirectly represses *ETT/ARF3* via the tasiR-ARF pathway. AS2–AS1 induce the accumulation of miR390 involved in the generation of tasiR-ARF. Subsequently, both the *ETT/ARF3* and *ARF4* transcripts are degraded (Figure 2b). Therefore, the AS2–AS1 complex represses the expression of *ETT/ARF3* in the dual pathway [49]. Several phenotypes in *as2*, including defects of development along the adaxial–abaxial axis, are suppressed by the *ett arf4* double mutations. Consistent with these results, an overexpression of a tasiR-ARF-insensitive ETT/ARF3 cDNA produces *as2*-like leaves [51]. Similarly, lamina phenotypes of *as1* are also suppressed by the *ett arf4* double mutation. These results suggest that the elevated *ETT/ARF3* and *ARF4* expression in *as2* and *as1* cause several leaf phenotypes, including defects of adaxial–abaxial polarity in these mutants. The importance of the repression of these *ARFs* by AS2–AS1 is further confirmed by the analysis of modifier mutations of *as2* and *as1*, which are described in the next section. Increased expression levels of *KAN2* and *YAB5* in *as2* and *as1* are caused by indirect regulation by AS2–AS1 [49].

## 3. Modifier Mutations That Enhance Defects of *AS2* and *AS1* in Leaf Adaxial–Abaxial Polarity

Various mutations (about 60) that markedly enhance the defects of adaxial leaf development in *as2* or *as1* have been reported [2]. The genes responsible for these mutations are considered as “modifiers” or modifier genes, which affect the phenotypic expression of other genes. Double mutants generate abaxialized filamentous (needle-like, pin-shaped, pointed) leaves that have lost the adaxial domain (Figure 3). Causative mutations occur in genes that are involved in chromatin modification, biogenesis of small RNAs, and DNA replication [2,52]. Mutations in genes encoding ribosomal proteins are also identified as modifiers in *as2* or *as1* [2,53,54,55,56,57,58]. In addition, mutations in genes encoding nucleolar proteins, such as *RNA HELICASE10* (*RH10*), *NUCLEOLIN1* (*NUC1*), *ROOT INITIATION DEFECTIVE2* (*RID2*), and *APUM23* are involved in ribosome biosynthesis, and enhance the phenotypes of *as2* and *as1* [59,60,61,62,63]. Mutations in *HDT1* and *HDT2* for nucleolar histone deacetylases (HDACs), which localize to the nucleolus, also act as modifiers of the *as2* and *as1* phenotypes [35]. We especially focus on the roles of nucleolar proteins in this review (Table 1).

Transcript levels of several abaxial-determining genes (*KAN2*, *YAB5*, *ETT/ARF3*, and *ARF4*) are slightly upregulated in the *as2-1* single mutant and each of the modifier single mutants and are markedly increased in the *as2-1* and modifier double mutants (for example, *as2-1 rh10-1*). When the double mutations of *ETT/ARF3* and *ARF4* are introduced to double mutants with *as2-1* and one of the modifier mutations, such as *as2-1 nuc1-1* or *as2-1 rh10-1*, the abaxialized filamentous leaves phenotype (e.g., *as2 rh10* leaves in Figure 3) is restored to the expanded shapes [59,64,65]. These results show that the upregulation of these *ARF* genes in the double mutants is responsible for the disappearance of their adaxial specification in their filamentous leaves. These genetic observations suggest that the repression of these *ARF* genes by the synergistic action of AS2–AS1 and modifier proteins is critical for the proper development of the adaxial domain. These results suggest that modifier proteins act cooperatively with AS2–AS1 to generate flat and symmetric leaves (Figure 3). The modifier genes that encode nucleolar proteins are summarized below.

Nucleoli are membrane-less organelles that appear to assemble through the phase separation of their molecular components [66]. The nucleoli contain internal subcompartments of ribosome biogenesis such as rDNA transcription, the processing of the precursor rRNA to generate mature rRNAs, assembly of these rRNAs, and many ribosomal proteins to generate each of small and large subunits of ribosomes. Genomic regions positioned in close proximity to the nucleolus are known as nucleolus-associated domains (NADs). Recent analyses of DNA sequencing that have been purified along with the nucleolus suggests that NADs in both animal and plant cells are enriched in regions displaying heterochromatic signatures [67,68].

***NUCLEOLIN1* (*NUC1*) gene**: Nucleolin, one of the most abundant non-ribosomal proteins in the nucleolus, has been described in a large variety of organisms [69]. The Arabidopsis genome encodes two nucleolin-like proteins—NUC1 and NUC2. Only the NUC1 gene, however, is ubiquitously expressed under normal growth conditions [61].

The single mutant *nuc1-1* exhibits a pointed narrow leaf shape, which is often observed in other modifier mutations [59,61,70,71]. In *nuc1-1* plants, nucleolar disorganization is observed and accumulated levels of pre-rRNA precursors are detected, indicating that NUC1 is involved in the processing of pre-rRNAs [61,72,73,74]. An analysis of high-throughput sequencing of DNA purified from the nucleoli of the *NUC1* mutant revealed that NUC1 is required for global genomic organization and stability [67,75]. In addition, human nucleolin is reported to be an assembly intermediate of the SSUP and its candidate components [62,76,77]. The *as2-1 nuc1-1* and *as1-1 nuc1-1* double mutant plants generate filamentous leaves. These mutant phenotypes are partially suppressed by the mutation in *ETT/ARF3*, indicating a role in the repression of *ETT/ARF3* gene expression for the formation of flat symmetric leaves in the wild-type plants [65].

***RNA HELICASE10* (*RH10*) gene**: The mutation of *rh10* was isolated as a modifier of *as2* and *as1*. Transcript levels of the abaxial genes, such as *ETT/ARF3* and *ARF4*, are elevated in *as2-1 rh10-1*, generating abaxialized filamentous leaves. This phenotype is suppressed by the *ett/arf3 arf4* double mutation, indicating a role in the repression of *ETT/ARF3* and *ARF4* gene expression for the formation of flat symmetric leaves in the wild-type plants [59]. RH10 is localized to the nucleolus in leaf primordia cells and is an ortholog of budding yeast Rrp3 and human DDX47, which belong to the DEAD-box RNA helicase family, a component of the nucleolar protein complex designated as the small subunit (SSUP) involved in 18S rRNA biogenesis [77,78]. It is reported that the DEAD-box RNA helicase family has an indispensable role in gene regulation through RNA metabolism [77,78,79,80]. DDX47 is necessary for maintaining the pluripotency of mouse stem cells [81]. In *rh10-1*, various defects are detected in SSUP-related events, such as the accumulation of 35S/33S rRNA precursors and a reduction in the 18S/25S ratio [59]. Nucleoli are enlarged in the *rh10-1* mutant [59]. RH10 may be involved in the early stages of processing reactions of the precursors of ribosomal RNAs.

***ROOT INITIATION DEFECTIVE2* (*RID2*) gene**: *RID2* encodes an evolutionarily conserved methyltransferase-like protein, an orthologous protein of the budding yeast, Bud23, which exhibits tight functional and physical interactions with some of the SSUP components [82,83,84]. The RID2 protein is localized in nuclei and accumulates mainly within nucleoli [60]. RID2 is involved in the processing of pre-rRNAs at various early stages [85,86]. Nucleolar enlargement is also observed in the *rid2* mutant. A mutation in the *RID2* gene has an effect on the adaxial–abaxial organization of leaves on the *as2* background, generating filamentous leaves and upregulating *ARF3/ETT* and *ARF4* as found in other modifier mutants and *as2-1* [59].

***APUM23* gene**: *APUM23*, which encodes a protein that is a member of the Pumilio/PUF domain protein family with its pumilio-like RNA-binding repeats, is localized to the nucleolus and is involved in the processing of 35S pre-rRNA [63,87]. The *apum23-1* mutant has enlarged nucleoli [63]. The double mutants *apum23-3 as2-2* and *apum23-3 as1-1* produce filamentous leaves, suggestive of the involvement of APUM23 in leaf development, similarly as with other nucleolar modifiers.

**Ribosomal protein genes**: It is worth noting that Arabidopsis double mutants of the ribosomal protein gene *rps6a-1*, which has a 9 bp deletion in the coding region of *RPS6A*, and *as2-1* exhibit strong adaxial leaf defects, as indicated by the fact that 80% of the double mutant leaves are filamentous [56]. Rps6 of budding yeast is one of five small-ribosomal-subunit proteins (Rps4, Rps6, Rps7, Rps9, and Rps14) that are components of the SSUP, which is a large ribonucleoprotein required for the biogenesis of the 18S rRNA [88]. Genetic interactions between *AS2/AS1* and homologues of *Rps4*, *Rps7*, *Rps9*, and Rps14 in Arabidopsis have yet to be examined. Three other *RPS* mutants (*rps23B*, *rps23B*, and *rps23B*) and fourteen *RPL* genes for ribosomal proteins in the large subunit also enhance the leaf phenotype in *as2* and/or *as1* (Table 1). It would also be intriguing to examine the relationships between the adaxial defects and ribosomal protein genes for such ribonucleoprotein complexes as a large subunit processome [89] in the nucleolus of Arabidopsis [53,54,55,56,57]. Therefore, the wild-type *AS2* gene, which is specific in plants, may appear to attenuate defects resulting from mutations in the ribosomal protein gene.

***HDT1* and *HDT2* genes**: *HDT1*, which encodes plant-specific nucleolar histone deacetylases (HDACs), is one of the factors responsible for gene silencing of megabase-scale rRNA loci and gene dosage control in nucleolar dominance [90,91], which are achieved by a highly condensed heterochromatic state that is associated with H3K9me2 and 5-methylcytosine enrichment in the promoter regions of rDNA genes [91]. Knockdown of the Arabidopsis genes *HDT1* and *HDT2* for nucleolar histone deacetylases (HDACs) enhances the leaf adaxial defects of *as2* and *as1* to generate severely abaxialized filamentous leaves, as seen in *as2-1 rh10-1* [35]. Considering the role of HDT1 in an epigenetic silencing of rDNAs (in nucleolar dominance), such as in the allopolyploid hybrid *Arabidopsis suecica* between *A. thaliana* and *A. arenosa* [91], the cooperative repression of the abaxial genes by AS2 and epigenetic silencing system of rDNAs described above are involved in the development of flat symmetric leaves.

## 4. AS2 Bodies: Perinucleolar Granules Co-Localized Partially with the Chromocenter

The AS2-fused YFP (Yellow Fluorescent Protein) was used to investigate subnuclear localization of AS2 protein. The AS2 protein is localized to perinucleolar bodies known as AS2 bodies as well as to the nucleoplasm in the leaf cells of Arabidopsis and some interphase cells of a cultured tobacco cell line BY-2 (Figure 4) [35,93]. As mentioned in Section 2, AS2 has the AS2/LOB domain that includes ZF, IcG, and LZL regions (Figure 2a), which are essential for the formation of AS2 bodies at the perinucleolar regions [94]; the carboxyl-terminal half of AS2 is nonessential for the body formation, but essential for the developmental function of AS2 [12,93]; AS1 co-localizes with AS2 in the cell bodies (Figure 4) [35].

In addition, the amino acid residues that are highly conserved within and adjacent to the ZF motifs of all the AS2 family members are critically important for the body formation: four cysteine residues; proline and alanine residues next to the first cysteine residue; the RRK cluster (Figure 2). The RRK sequence is found within proposed nucleolar localization signals (NoLSs) [95,96,97,98] and it is likely that this cluster participates in the perinucleolar localization of AS2. These amino acid residues and three regions (ZF, IcG, and LZL) in the AS2/LOB domain are also required for the ability of AS2 to complement the *as2* mutation and to bind to the coding sequence of the target *ARF3/ETT* gene, showing that the formation of AS2 bodies is related to the genetic functions of AS2 in leaf formation. The AS2 bodies appear to be located to the peripheral regions of nucleoli and are partially overlapped with perinucleolar chromocenters with condensed chromatin-containing ribosomal RNA genes (45S rDNA repeats), suggesting that AS2 bodies interact with 45S rDNA repeats (Figure 4) [94].

It should be noted that the proportions of cells in which AS2 bodies are generated in plants differ from those in cultured cells. AS2 bodies are detected in only a few percentages of interphase cells of the tobacco-cultured cell line BY-2 and the Arabidopsis-cultured cell line MM2d transformed with the AS2-fused YFP constructs [93,94]. AS2 bodies are, however, detected in almost all interphase cells of the adaxial domain in leaf primordia of the Arabidopsis plants with the AS2-fused YFP construct [94]. The average number of AS2 bodies per YFP-positive cells at interphase (and/or the G0 stage) in leaf primordia was calculated as 1.9 [94]. In contrast, AS2 bodies are formed in all M phase cells of both cell lines, MM2d and BY-2 and in all M phase cells of leaf primordia; AS2 bodies are separated into daughter cells during the M phase progression [93,94]. These observations imply that the formation and distribution of AS2 bodies might be modulated developmentally in plants and in a cell-cycle-dependent manner.

The subcellular localization of AS2 appears to be subject to multiple controls, since AS2 was exported to the cytoplasm via the action of the geminivirus-encoded nuclear shuttle protein [12,94,99].

Although mechanisms for the formation of AS2 bodies and their roles in repressing the target genes for leaf development have yet to be discovered, the identification of AS2 body components and investigations of how these molecules interact within the nucleolus would provide answers for these questions.

## 5. AS2–AS1 Binds to Exon 1 of the Target Gene *ETT/ARF3*, and Is Involved in Maintaining CpG Methylation in Exon 6

Four mechanisms have been investigated for the repression of target gene *ETT/ARF3* expression by AS2–AS1: (1) direct binding of the AS1–AS2 complex to the 5′-upstream regions of *ETT/ARF3* to reduce the expression activity of *ETT/ARF3* (Figure 5) [49]; (2) indirect activation of miR390-dependent post-transcriptional gene silencing to negatively regulate both *ETT/ARF3* and *ARF4* (Figure 2b) [49,50]; (3) direct binding of AS2 to the synthetic GCGGCG-containing nucleotides [47,50], and exon 1 of the *ETT/ARF3* gene containing the CGCCGC (Figure 5) [65]; (4) maintenance of the status of gene body (CpG) methylation in exon 6 of ETT/ARF3 (Figure 5) [49,65,100]. In the present review, we focus on the last two topics.

Several protein members of the AS2/LOB family, including AS2, bind synthetic double-stranded DNAs containing the GCGGCG sequence [101]. AS2, specifically, also binds in vitro the double-stranded CGCCGC sequence in exon 1 of the target gene *ETT/ARF3* [65]. The zinc-finger motif containing the RRK (Arg-Arg-Lys) sequence in AS2 is essential for this binding [65], the formation of AS2 bodies and functions in the development of leaves with normal shapes [94]. Modes of molecular interactions between the amino acid residues in RRK and each of the deoxyribonucleotides in GCGGCG have recently been proposed based on the results of SEC–SAXS (size exclusion chromatography–small angle X-ray scattering) experiments [47]. Since 32 out of 42 members of the family harbor the RRK and/or RRR sequence in the ZF motifs [12], it should be informative to investigate the possible roles of the clusters of these basic amino acid residues in other members in plant physiology, development, and growth [102].

AS2 and AS1 play a role in maintaining cytosine methylation mediated by METHYLTRANSFERASE1 (MET1) in six CpG dinucleotides in exon 6 of *ETT/ARF3* (Figure 5) [49]. Because levels of CpG methylation are inversely related to the *ETT/ARF3* transcript levels, AS2 and AS1 possibly regulate the transcriptional repression of *ETT/ARF3* through CpG methylation in the recruitment of methylation activity and/or inhibition of demethylation activity at exon 6 [103]. As described in Section 3, mutations in the *RH10*, *NUC1*, and *RID2* genes for nucleolar proteins enhance defects in leaf morphology in the *as2* mutant and, in parallel with this observation, result in an increase in the transcript level of target genes *ETT/ARF3* and *ARF4*. The levels of CpG methylation at some of the CpG dinucleotides in exon 6 of *ETT/ARF3* decrease in *rh10* and *nuc1* mutants, and further decrease in *rh10 as2* and *nuc1 as2*, suggesting that these nucleolar proteins, in addition to AS2, also take part in maintaining the cytosine methylation of CpG dinucleotides in exon 6 of *ETT/ARF3* [65,103].

How can AS2 be involved in maintaining MET1-regulated CpG methylation in exon 6 of the *ETT/ARF3* gene? MET1 is an ortholog of the Dnmt1 of vertebrates and acts as DNA methyltransferase, which methylates hemimethylated CpG, converting it to fully methylated CpG during DNA replication [104,105]. MET1 is part of a putative protein complex involved in the maintenance of DNA methylation in Arabidopsis [106,107,108,109,110,111]. MET1 is similar to Dnmt1, in terms of the domain organization [109,112], except that MET1 has no amino acid sequence for the ZF-CxxC motif. If AS2 forms a protein complex with MET1, AS2 provides the ZF motif, which has DNA binding activity, as described above in this section, to the MET1-containing putative protein complex. The promoter regions of inactive 45S rDNAs in Arabidopsis are highly methylated by MET1 and their chromatin states are highly condensed at perinucleolar regions [113,114]. MET1 requires NUC1, one of the AS2 modifiers, and nucleolar histone deacetylase HDA6 for this methylation [113,114] and directly interacts with HDA6 [115,116], which is also associated with AS2 and AS1 [36]. The CpG methylation system for the 45S rDNA might be also involved in CpG methylation in the *ETT/ARF3* gene around perinucleolar areas; the *ETT/ARF3* gene might be recruited to such an area by an action of AS2 (Figure 6).

The AS2–AS1 complex is also involved in the establishment of leaf proximal–distal polarity to repress the class 1 *KNOX* homeobox genes *BREVIPEDICELLUS* (*BP*), *KNAT2*, and *KNAT6* (Figure 1) [18]. The AS2–AS1 complex physically interacts with CURLY LEAF (CLF), the polycomb repressive complex 2 (PRC2) core component, and LIKE-HETEROCHROMATIN PROTEIN1 (LHP1), the PRC1 component, and recruits PRC2 to the homeobox genes *BP* and *KNAT2* [117,118]. AS2–AS1 interacts with the *BP* promoter, likely through the recruitment of the chromatin-remodeling factor HIRA (histone-regulator A) and forms a repressive chromatin state [19]. AS2–AS1 also interacts with LEAF FLOWER RELATED (LFR) in the chromatin remodeling complex and is associated with H3K27me3 in the *BP* gene, but not with the *ETT/ARF3* gene [119]. AS2–AS1 is required for the correct temporal repression of *ETT/ARF3*, which involves a PRC2-independent mechanism [50]. Despite their pivotal role, the means by which AS2–AS1 epigenetically represses *ETT/ARF3* in the establishment of leaf adaxial–abaxial polarity remains unsolved.

## 6. Subcellular localization of AS2

Although it is often reported that AS2 and other members of the AS2/LOB family are nuclear proteins [2,12,35,38,93,120], mechanisms of the nuclear localization of AS2 protein are poorly understood. The RRK sequence in the zinc-finger motif of AS2 (Figure 2a) is only a basic amino acid cluster, which is thought to be critical for nuclear localization. The examination of subcellular localization of the mutant as2 (as2-RRK/3A) with the alanine replacement at the RRK sequence in the zinc-finger motif (Figure 2a) with the alanine cluster, however, shows that the mutant protein is still present in the nucleoplasm; it is not exported to the cytoplasm and does not form the AS2 bodies at the perinucleolar region [94]. These observations show that the RRK sequence is not involved in the nuclear localization of AS2. This result is consistent with the previous finding [121]: that is, the mutant proteins of ASL18/LBD16, another member of the AS2/LOB family, from which the RRK sequence is deleted are still localized to the nucleus and nuclear localization signals are proposed to be present in the coiled-coil sequence and the carboxyl-terminal region. Furthermore, the mutant as2 protein that lacks the IcG region is exclusively localized to the cytoplasm [94]. It is also reported that AS2 was exported to the cytoplasm via the action of the geminivirus-encoded nuclear shuttle protein and localized to the plant P-body complex [99]. Therefore, the AS2 protein might be subject to multiple subcellular localization controls, depending on its interactions with other proteins and other unknown cellular conditions.

As described above, 32 among the 42 members of the AS2/LOB family harbor ZF motifs, the amino acid sequences of which include RRK or RRR sequences [12,38]. The observation that the as2 mutant lacking RRK is localized to the nucleoplasm, but does not form AS2 bodies at the nucleolar periphery, suggests that RRK appears to be involved in the transport of AS2 to the perinucleolar region and/or to the formation of AS2 bodies. The RRK sequence is present within the proposed NoLSs [95,96,97,98]. It is intriguing to test whether this cluster of the basic amino acid residues in the zinc-finger could be directly involved in the transfer to the peripheral region of the nucleoli from the nucleoplasm and/or the formation of AS2 bodies by a phase separation mechanism, because the nucleolus and many nucleolar bodies are proposed to be formed through such a physico-chemical molecular interaction [122].

## 7. Possible Roles of AS2 Bodies in Epigenetic Repression of *ETT/ARF3*

As described in Section 3, the level of the leaf abaxial gene *ETT/ARF3* expression is influenced by modifier proteins that are localized to the nucleolus. For example, the AS2–AS1 complex binds directly to the upstream region of the *ETT/ARF3* gene to repress its transcription [49,64]. Furthermore, the *ETT/ARF3* transcriptional level is altered by mutations in various genes for nucleolus-localized proteins, such as RH10, RID2, and NUC1, which affect the biogenesis of ribosomal RNAs and the formation of the nucleolus with a properly organized morphology [59,60,61]. Perturbation of rRNA biogenesis correlates with structural disorders of the nucleolus, such as nucleolar enlargement in plant cellsand in animal cells [60,85,86,123]. It is, however, still unknown how structural disorders of the nucleolus affects leaf development mediated by AS2–AS1.

Perinucleolar regions might provide the molecular architectures, such as nucleolus-associated chromatin domains (NADs), which correspond to regions of low transcriptional levels [67,68,124,125]. In Arabidopsis, many of 45S rRNA genes are condensed as heterochromatin and silenced by epigenetic mechanisms that include DNA methylation and histone modification at the periphery of the nucleolus [114,126,127,128]. MET1, HDA6, and chromatin assembly factor (CAF-1) are all involved in the formation of such an epigenetic state in the perinucleolar subdomain [114,126,127]. AS2 and AS1, which are associated with HDA6 [36], are colocalized to AS2 bodies in the peripheral region of the nucleolus (Figure 4 and Figure 6) [35,93,94]. The *ETT/ARF3* gene undergoes MET1-dependent CpG methylation in exon 6 [49]. As described in Section 4 and Section 6, mutant proteins of AS2 (as2-RRK/3A, Figure 2a) that do not form AS2 bodies are not functional in leaf morphogenesis [94]. Mutations in RH10, RID2, and NUC1 might affect the integrity of nucleolar morphology [59,60,61], which would then alter the transcriptional patterns of *ETT/ARF3*, a target gene of AS2, although the subnucleolar localization of the target gene to the peripheral subdomain (Figure 6) has yet to be demonstrated. It is an interesting problem to elucidate how the *ETT/ARF3* gene transcribed by RNA polymerase II is regulated in the nucleolus or its peripheral region.

Recently, AS2 was shown to bind to DNAs other than *ETT/ARF3* [129]. Since AS2 is a plant-specific DNA binding protein, elucidation of the interaction between AS2 and these DNA molecules should uncover a more global and novel regulatory system mediated by AS2 and the nucleolus in plant cells [19,101].

## Figures and Tables

**Figure 1 ijms-21-07314-f001:**
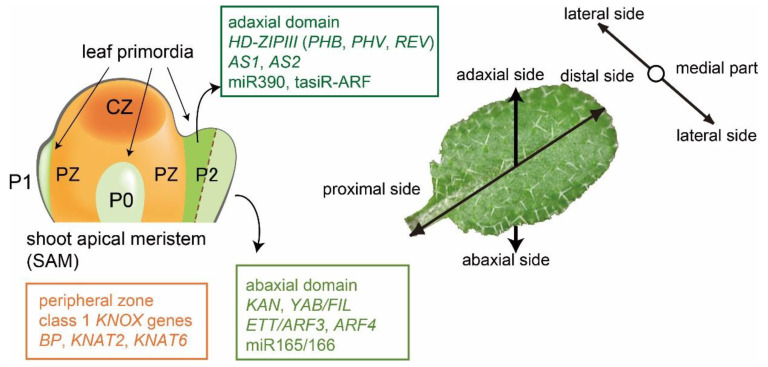
The leaf structure develops along three axes. Developmental compartments in the shoot apex around the apical meristem and the three structural leaf axes are schematically shown on the left and right sides, respectively (see details in text). CZ, central zone; PZ, peripheral zone; p0, primordium 0; p1, primordium 1; p2, primordium 2. Schematic representations are modified from ref. [2].

**Figure 2 ijms-21-07314-f002:**
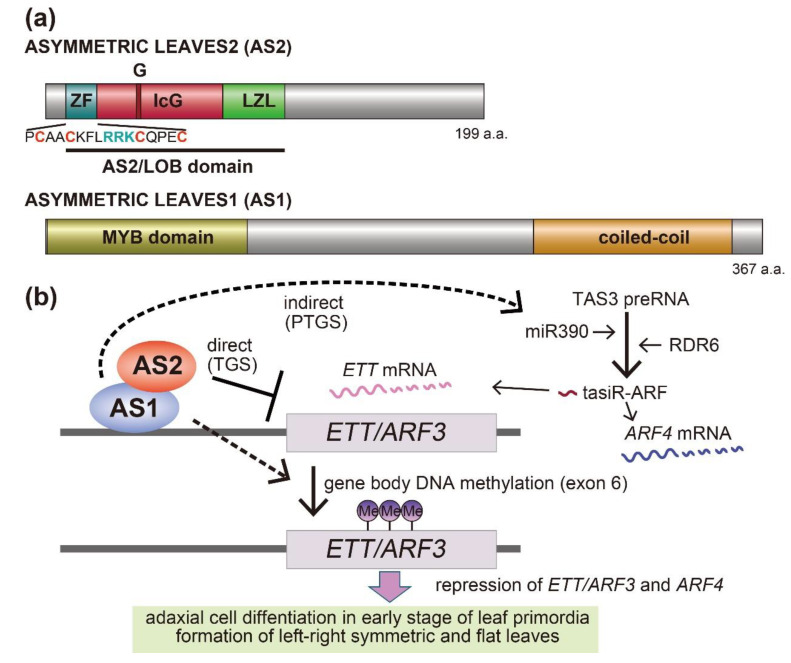
(**a**) Motif and domain organization of AS2 and AS1 proteins. The ZF motif, IcG, and LZL regions of AS2 and the MYB domain and coiled-coil structure of AS1 are shown. (**b**) Dual regulation of *ETT/ARF3* gene expression, including that by the possibly epigenetic system of AS2–AS1. The AS2–AS1 complex represses *ETT/ARF3* directly by binding to its promoter and represses *ETT/ARF3* and *ARF4* indirectly via stimulation of the miR390 and tasiR-ARF pathway. In addition, AS1 and AS2 maintain gene body DNA methylation of the *ETT/ARF3* gene. Solid lines indicate direct regulation and dashed black lines indicate indirect regulation. Schemes of (b) are modified from ref. [2].

**Figure 3 ijms-21-07314-f003:**
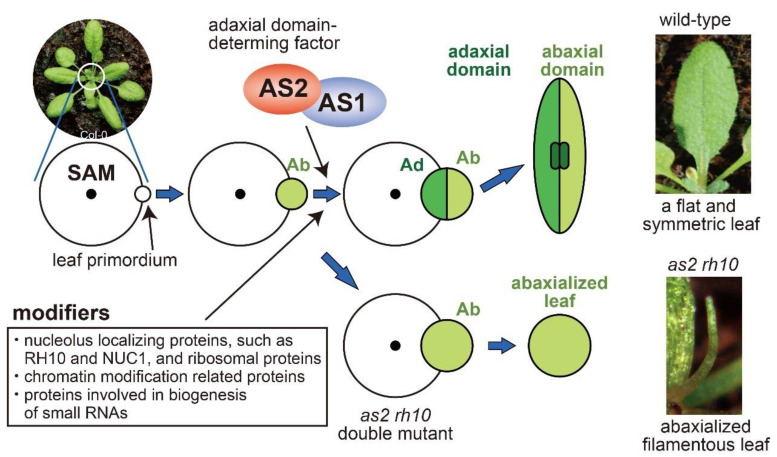
Development of leaves along with three axes. Top views of SAM are schematically shown by open circle. Dot indicates the center of the SAM. Adaxial and abaxial domains are shown by green and light green, respectively. AS2-AS1 contributes to the determination of adaxial domain followed by the medial-lateral growth of leaves with vasculature (indicated by dark green rectangles). Modifiers act cooperatively with AS2–AS1 at leaf primordia to develop the adaxial (Ad) domain from abaxialized (Ab) leaf primordia and to generate leaves with a flat and symmetric structure. The double mutation into AS2 (or AS1) and modifiers results in the production of abaxialized filamentous leaves. Photograph of wild-type leaf is modified from ref. [64].

**Figure 4 ijms-21-07314-f004:**
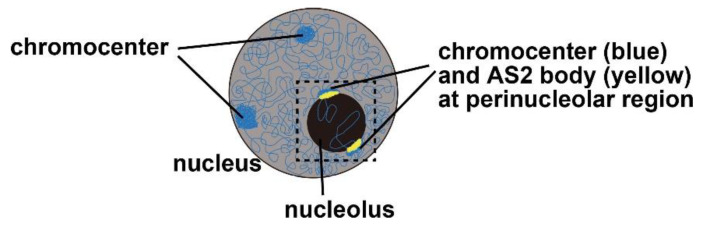
Schematic representation of the nucleus is shown. Chromosomes and AS2 bodies are indicated by blue and yellow, respectively. AS1 and AS2 are co-localized on AS2 bodies.

**Figure 5 ijms-21-07314-f005:**
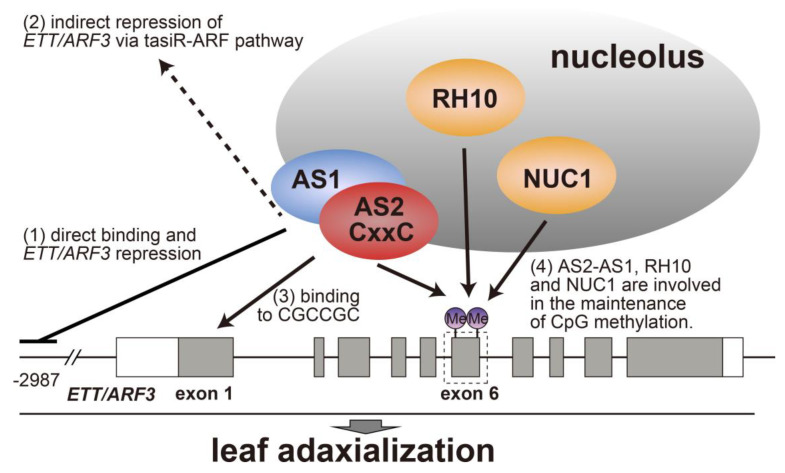
Molecular relationships between AS2–AS1 and the target gene *ETT/ARF3*. AS1–AS2 directly binds the *ETT/ARF3* regulatory region and represses *ETT/ARF3* expression. AS2 binds to the specific CGC repeat sequence in exon 1. AS2–AS1, RH10, and NUC1 are involved in the maintenance of CpG methylation in exon 6. RH10 and NUC1 are proteins localized in the nucleolus.

**Figure 6 ijms-21-07314-f006:**
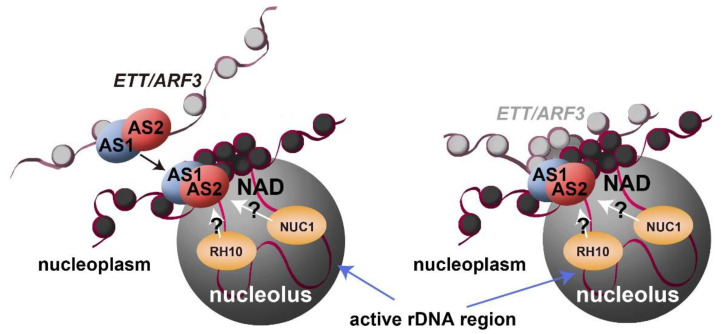
Models of the roles of nucleolar proteins in AS2–AS1 involved in epigenetic regulation of the *ETT/ARF3* gene. Nucleolus and surrounding structures are shown. Nucleosomes are indicated by coiled structure composed of red lines (DNA) and light/dark grey circle (histone octamer). Different genomes are distinguished by different darkness of nucleosomes. NUC1 affected the localization patterns of AS2 bodies at the peripheral region of the nucleolus, which are required for leaf development. AS2 bodies are partially overlapped with chromocenters, represented by dense nucleosome at the peripheral region of the nucleolus. The 45S rDNA repeat loci include transcriptionally active and inactive regions, which chromosomal status are loosened in the nucleolus and condensed on the peri-nucleolus (overlapping with chromocenter), respectively, suggestive of an interaction of AS2 bodies with inactive 45S rDNA. RH10 and NUC1 in addition to AS2 are involved in the maintenance of CpG methylation in exon 6 of *ETT/ARF3* in the nucleoplasm (left panel) or in AS2 bodies (right panel). NUC1 is involved in the maintenance of CpG methylation in 45S rDNA.

**Table 1 ijms-21-07314-t001:** Gene mutations that act as modifiers to enhance leaf adaxial–abaxial abnormalities in *as2* and *as1*.

1. Gene Name (Mutant Name)	2. AGI Code	3. Protein	4. Cellular Process and Status	5.References
I. Genes involved in rRNA processing				
*NUCLEOLIN1* (*nuc1*)	AT1G48920	NUCLEOLIN	rRNA processing and ribosome biogenesis Components of SSUP-like complex	[59,61,70,71]
*RNA HELICASE10* (*rh10*)	AT5G60990	DEAD-box RNA helicase family protein	pre-rRNA processing Components of SSUP-like complex	[59]
*ROOT INITIATION DEFECTIVE2* (*rid2*)	AT5G57280	RNA methyltransferase-like protein	pre-rRNA processing	[59,60]
*APUM23* (*apum23*)	AT1G72320	Pumillio protein containing PUF domain	pre-rRNA processing and rRNA maturation	[63]
II. Genes for ribosomal proteins				
*RPL4D (rpl4d)*	AT5G02870	Ribosomal proteins	Subunits of ribosome; components of pre-rRNA-protein complex	[53,54,55,56,57]
*RPL5A (pgy3/ae6/oli5/rpl5a)*	AT3G25520			
*RPL5B (rpl5b/oli7)*	AT5G39740			
*RPL7B (rpl7b)*	AT2G01250			
*RPL9c (rpl9c/pgy2)*	AT1G33140			
*RPL10aB (rpl10ab/pgy1)*	AT2G27530			
*RPL18C (rpl18c)*	AT5G27850			
*RPL24b (stv1)*	AT3G53020			
*RPL27ac (rpl27ac)*	AT1G70600			
*RPL28A (ae5/rpl28a)*	AT2G19730			
*PRL36aB (api2)*	AT4G14320			
*RPL36aA (rpl36aa)*	AT3G59540			
*RPL38B (rpl38b)*	AT4G31985			
*RPL39C (rpl39c)*	AT3G23390			
*RPS6A * (rps6a)*	AT4G31700			
*RPS21B (rps21b)*	AT3G53890			
*RPS24B (rps24b)*	AT5G28060			
*RPS28B (rps28b)*	AT5G03850			
III. Genes involved in histone modification				
*HDT1* (*hdt1*/*hd2a*/*hda3*)	AT3G44750	Histone deacetylase (plant-specific class)	Deacetylation of nucleosomal histone H3, transcription of rDNAs	[35,90,92]
*HDT2* (*hdt2*/*hd2b*)	AT5G22650	Histone deacetylase (plant-specific class)	Deacetylation of nucleosomal histone H3, transcription of rDNAs	[35,90,92]

* Rps6 of budding yeast is one of the proteins that was identified as a bona fide component of the SSUP.

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
