# Peer review of "Roles of ASYMMETRIC LEAVES2 (AS2) and Nucleolar Proteins in the Adaxial–Abaxial Polarity Specification at the Perinucleolar Region in Arabidopsis"

_ijms, 2020, doi:10.3390/ijms21197314_

Round 1

Reviewer 1 Report

In this review paper, the authors investigate the possible role of nucleolar proteins to control, together with specific Asymetric leaves1 and 2 the adaxial-abaxial polarity in leaves. The article is very well written, easy to read (beside the complexity of certain mechanisms) and contains 6 high quality figures and 1 table. A total of 110 references has been used in this review.

The authors have gathered recent studies and propose nice regulation mechanisms and regulatory models.

I have 3 main concerns about the paper and several minor suggestions.

1-      The authors often mention formation of an AS1-AS2 complex. This complex is not really described, nor the way in which it was determined (protein protein interaction sudies, co-localisation, 2hybrid analyses?) and this notion of complex should be better explained probably in the beginning of chapter 2.

  • I would recommend to introduce a small paragraph on the function and known roles of the nucleolus just before the description of the nucleolar proteins described on page 4 line 134. This is perhaps very well known for the nucleolus specialists but a short reminder would is probably necessary in this review for more basic reading audience, especially because the proposed role in this review is not exactly on the same line as the more known rRNA maturation function of the nucleolus.

3-      Construction of the chapter 4 is to my point of view not adapted: chapter 4 is reviewing localization of the AS proteins in perinucleolar bodies. It is largely built (along with others) on a very recent (2020) article of which 3 of the authors are themselves authors, which is not a problem in itself, but which is too close to a results article without giving the methods or the discussion.  Probably because the authors are particularly familiar with this work, the reader do not know in its actual form if fig 4 presents non published data or a figure reproduced from an article (apparently not when consulting ref #73) and the way it is built makes it difficult to understand, at least in a review article. Reading the lines 214 to 222, for example, does make it possible to know whether they are data already published or not in this form. It would undoubtedly be simpler to make a summary drawing of fig. 4 (as fig 6 for example) to better explain/indicate the co-localization of the proteins of the process as well as the protein domains involved, with citations to the original articles

Minor comments

  • Line 73: are the expression patterns of AS1 and AS2 matching with putative constitution of a protein complex?
  • Line 75: “inner domain”, please specify
  • Line 79: “42 members”, in Ath?
  • Line 83: it is not clear if other genes with the same function are not in this family or if AS2 is the only one to have this function in the family
  • Line 111: please give some details about modifier mutations. This has not really been explained.
  • Line 126: filamentous leaves. Please announce figure 3 for pictures.
  • Line 205: explain or discuss “modifiers act cooperatively”
  • Line 234 is highly speculative
  • Chapter 5: begins with 4 mechanisms for the repression of target gene ATT/ARF3. I would suggest to use the same numbering on figure 5 (on the arrows) with addition of mechanism 2 (activation of miR390) which is not present on figure 5 yet. Thus, authors could combine figure 2-b with figure 5 to have a more complete figure without actual redundancy.

Author Response

We are grateful to Reviewer 1 for the critical comments and useful suggestions that have greatly helped us to improved our paper. As indicated in the responses that follow, we have taken all the comments and suggestions into account in the revised version of our manuscript.

<Comments of Reviewer 1>

<Comment #1>: The authors often mention formation of an AS1-AS2 complex. This complex is not really described, nor the way in which it was determined (protein protein interaction sudies, co-localisation, 2hybrid analyses?) and this notion of complex should be better explained probably in the beginning of chapter 2.

<Response>: We added the sentences that describe the AS2-AS1 complex.

Lines 89-95: As described above, AS2 and AS1 proteins, which have AS2/LOB and R2R3 MYB (SANT) domains, respectively (Figure 2a), are identified by yeast two-hybrid system, pull-down and gel-shift assays, and subcellular co-localization analyses. These experiments because of their nature indicate that AS2 and AS1 are physically associated with each other in vitro [19,33-36], implying also that they form a protein complex in the nucleus. Transcripts of AS2 and AS1 genes accumulate throughout the entire leaf primordia at early stages, in which the AS2-AS1 complex might be formed, but the accumulation patterns change as the leaves develop [37].

<Comment #2>: I would recommend to introduce a small paragraph on the function and known roles of the nucleolus just before the description of the nucleolar proteins described on page 4 line 134. This is perhaps very well known for the nucleolus specialists but a short reminder would is probably necessary in this review for more basic reading audience, especially because the proposed role in this review is not exactly on the same line as the more known rRNA maturation function of the nucleolus.

<Response>: We added the sentences that introduce the function and known roles of the nucleolus.

Lines 164-171:

Nucleoli are membrane-less organelles that appear to assemble through phase separation of their molecular components [66]. The nucleoli contain internal sub-compartments of ribosome biogenesis such as rDNA transcription, processing of the precursor rRNA to generate mature rRNAs, assembly of these rRNAs, and many ribosomal proteins to generate each of small- and large-subunits of ribosomes. Genomic regions positioned in close proximity to the nucleolus are known as nucleolus-associated domains (NADs). Recent analyses of DNA sequencing that have been purified along with the nucleolus suggests that NADs in both animal and plant cells are enriched in regions displaying heterochromatic signatures [67,68].

<Comment #3>: Construction of the chapter 4 is to my point of view not adapted: chapter 4 is reviewing localization of the AS proteins in perinucleolar bodies. It is largely built (along with others) on a very recent (2020) article of which 3 of the authors are themselves authors, which is not a problem in itself, but which is too close to a results article without giving the methods or the discussion.  Probably because the authors are particularly familiar with this work, the reader do not know in its actual form if fig 4 presents non published data or a figure reproduced from an article (apparently not when consulting ref #73) and the way it is built makes it difficult to understand, at least in a review article. Reading the lines 214 to 222, for example, does make it possible to know whether they are data already published or not in this form. It would undoubtedly be simpler to make a summary drawing of fig. 4 (as fig 6 for example) to better explain/indicate the co-localization of the proteins of the process as well as the protein domains involved, with citations to the original articles

<Response>: We have taken the comments and suggestions and we changed Figure 4.

We removed the old figure 4 that contained pictures with a confocal laser scanning microscope. We moved upper panel of old Figure 6 to new Figure 4 to make a summary drawing. We think that Figure 4 becomes better to explain the co-localization of the proteins of the process as well as the protein domains involved. We cited the original articles (12, 35, 93, 94, 99) in the chapter 4.

<Minor comments of Reviewer 1>

<Minor comment #1>: Line 73: are the expression patterns of AS1 and AS2 matching with putative constitution of a protein complex?

<Response>: The expression patterns of AS1 and AS2 at early stage of leaf primordia match with putative constitution of a protein complex, but expression patterns of AS1 and AS2 at later stage of leaf primordia are different. Therefore, we focus the common function of AS1 and AS2 in this review. Specially, we focus the molecular function of AS1 and AS2 for repression of the ETT/ARF3 gene. We added the sentences in lines 89-95, as reviewer 1 suggested, in which  we carefully described.  

<Minor comment #2>: Line 75: “inner domain”, please specify

<Response>: Since “inner domain” is not clearly specified, we described as follows.

Line 96-97: while AS1 transcripts are detected in the central region between the adaxial and abaxial domains of leaf primordia and the vasculature regions in more developed primordia (10, 37).

<Minor comment #3>: Line 79: “42 members”, in Ath?

<Response>: Line 101: We added “in Arabidopsis”

<Minor comment #4>: Line 83: it is not clear if other genes with the same function are not in this family or if AS2 is the only one to have this function in the family

<Response>: We changed and added the sentences as follows.

Lines 103-110:

Members that might retain functions similar to those of the AS2 gene do not appear to exist in this family, because substitution of the AS2/LOB domain of AS2 with those of other members disrupts its function [39]. Considering similarities among the AS2/LOB domains, it is, however, undeniable that these family members may retain partially overlapping functions at the molecular level. They have often been described as transcription factors [40-47]. Recent results on AS2, however, suggest that the term “transcription factor” is not appropriate for a member of this family; and they are better described as novel functional factors that could play a role in gene expression.

Lines 33-34: and predict that AS2 could be a molecule with novel functions that cannot be explained by the conventional concept of transcription factors.

<Minor comment #5>: Line 111: please give some details about modifier mutations. This has not really been explained.

<Response>: We added the sentences as follows.

Lines 140-142: Various mutations (about 60) that markedly enhance the defects of adaxial leaf development in as2 or as1 have been reported [2]. The genes responsible for these mutations are considered as "modifiers" or modifier genes, which affect the phenotypic expression of other genes.

<Minor comment #6>: Line 126: filamentous leaves. Please announce figure 3 for pictures.

<Response>: We added “e.g. as2 rh10 leaves in figure 3” in line 157

<Minor comment #7>: Line 205: explain or discuss “modifiers act cooperatively”

<Response>: We added the sentences as follows.

Lines 161-162: These results indicate that modifier genes act cooperatively with AS2-AS1 to generate a flat and symmetric leaves (Figure 3). 

<Minor comment #8>: Line 234 is highly speculative

<Response>: We removed the sentence, “There might be correlation between the role of AS2 bodies, epigenetic repression of the target gene ETT/ARF3, and chromatin condensation of 45S rDNA repeats”, because we also think that it is highly speculative.

<Minor comment #9>: Chapter 5: begins with 4 mechanisms for the repression of target gene ATT/ARF3. I would suggest to use the same numbering on figure 5 (on the arrows) with addition of mechanism 2 (activation of miR390) which is not present on figure 5 yet. Thus, authors could combine figure 2-b with figure 5 to have a more complete figure without actual redundancy.

<Response>: We added the same numbering on Figure 5 (on the arrows) with addition of mechanism 2 (activation of miR390).

Reviewer 2 Report

The authors concisely summarized the molecular functions of ASYMMETRIC LEAVES2 (AS2) a key factor of leaf polarity regulations in this review. Most part is self-introductory, but considering that the authors have lead front lines of studies on molecular roles of AS2 after the molecular cloning of this gene, this is very natural. Figures are well designed and easy to understand. Sections are also well organized. Thus I have only a few points to be fixed before publication of this review article.

First, role of YABBY genes in abaxial identification of leaves. As summarized in a paragraph of Lines 122-132, misexpression of ARFs seems to be the key reason why the as2 has a defect in the dorsiventral characterization. So upregulation of YAB5 might not be a key. In addition, as Sarojam et al. 2010 showed, the molecular role of the YABBY I is not in the abaxial identification, but in starting the leaf-lamina program with shut-off of SAM program. I think that YABBY I is not abaxial identity gene. Throughout this review, the authors are requested to be careful on the role of YABBY I (Line 54).  

Secondly, considering the historical knowledge on the misregulation of the class I KNOX (KNOX I) in the as1 and as2 mutant leaves, readers would like to know on how KNOX I regulation is important for the AS2 role. In this review the authors only briefly described on the repression of KNOX I by AS2 (Lines 311-313). In addition, the authors wrote that this repression is related to leaf proximal-distal polarity without any detailed explanation. Please add some explanations here. Some more references can be cited on this topic?

Minor comments:

  • 1. Orange-colored ‘peripheral zone’ box is now put under the black-lettered ‘leaf primordia’. This arrangement will make a misunderstanding (as like “peripheral of leaf primordia”), so I recommend the authors to exchange ‘shoot apical meristem (SAM) and ‘leaf primordia’ at their position in this figure.
  • Line 80 ‘plant individuals’: Is this required? I think that this word might cause some misunderstandings.
  • Line 135 ‘In contrast to animals and yeast’: how plants are different from them?
  • Lines 167 and 174 ‘Nucleolar enlargement’ and ‘enlarged nucleoli’: What can be deduced from this phenotype? Nucleolar/ribosomal stress symptom?
  • Line 379: We need reference on this sentence (recently, AS2 was shown…).

Author Response

We are grateful to Reviewer 2 for the critical comments and useful suggestions that have greatly helped us to improved our paper. As indicated in the responses that follow, we have taken all the comments and suggestions into account in the revised version of our manuscript.

<Comments of Reviewer 2>

<Comment #1>: First, role of YABBY genes in abaxial identification of leaves. As summarized in a paragraph of Lines 122-132, misexpression of ARFs seems to be the key reason why the as2 has a defect in the dorsiventral characterization. So upregulation of YAB5 might not be a key. In addition, as Sarojam et al. 2010 showed, the molecular role of the YABBY I is not in the abaxial identification, but in starting the leaf-lamina program with shut-off of SAM program. I think that YABBY I is not abaxial identity gene. Throughout this review, the authors are requested to be careful on the role of YABBY I (Line 54).  

<Response>: We added the sentences as follows.

Lines 67-74:

Members of the KANADI (KAN) gene family, which encode proteins with the GARP domain, determine abaxial cell fate [22,25]. The Arabidopsis genome contains six YABBY genes, which encode transcription factors with a zinc finger domain and an HMG-related domain with a helix-loop-helix structure. The three YABBY genes, FILAMENTOUS FLOWER (FIL), YABBY3 (YAB3), and YAB2 express in the abaxial domains of all leaf-derived organs, including cotyledons, leaves, and floral organs [26-30]. Furthermore, genetic analyses have shown that four YABBY genes (FIL, YAB3, YAB2, YAB5) govern embryo patterning and growth of leaf lamina along the abaxial–adaxial boundary.

<Comment #2>: Secondly, considering the historical knowledge on the misregulation of the class I KNOX (KNOX I) in the as1 and as2 mutant leaves, readers would like to know on how KNOX I regulation is important for the AS2 role. In this review the authors only briefly described on the repression of KNOX I by AS2 (Lines 311-313). In addition, the authors wrote that this repression is related to leaf proximal-distal polarity without any detailed explanation. Please add some explanations here. Some more references can be cited on this topic?

<Response>: The class 1 KNOX regulation by AS1 and AS2 is important for leaf development. Therefore, we have added the sentences as follows in chapter 1.

Lines 50-62:

The Rough Sheath2 (RS2) gene of maize, an ortholog of PHANTASTICA (PHAN) of Antirrhinum majus and AS1 of Arabidopsis, is involved in the proximal-distal patterning of maize leaves through repression of class 1 KNOX genes [10,14,15]. The PHAN gene is involved in growth and the adaxial-abaxial determination of lateral organs. In addition, its activity is required early in the growth of leaves in the direction of the proximal-distal axis [16,17]. The ectopic expression of class 1 KNOX genes in as1 and as2 mutant plants results in reductions in the growth of leaf blades and petioles in Arabidopsis, and these phenotypes are suppressed by mutations of the class 1 KNOX genes, brevipedicellus (bp), knat2, and knat6. These results indicate that the AS1 and AS2 genes of Arabidopsis are involved in the establishment of the proximal-distal axis through repression of the class 1 KNOX genes [18]. In addition, the formation of shorter petioles and leaf blades in as1 and as2 is due to repression of GA-synthetic genes by the upregulation of BP/KNAT1, KNAT2, and KNAT6 [18]. AS1, acting together with AS2, directly represses the expression of the BP and KNAT2 genes [19]. In this review, we focus on the establishment of leaf adaxial-abaxial polarity.

<Minor comments of Reviewer 2>

<Minor comment #1>: Orange-colored ‘peripheral zone’ box is now put under the black-lettered ‘leaf primordia’. This arrangement will make a misunderstanding (as like “peripheral of leaf primordia”), so I recommend the authors to exchange ‘shoot apical meristem (SAM) and ‘leaf primordia’ at their position in this figure.

<Response>: Thank you for your suggestion. We exchanged ‘shoot apical meristem (SAM) and ‘leaf primordia’ at their position in Figure 1.

<Minor comment #2>: Line 80 ‘plant individuals’: Is this required? I think that this word might cause some misunderstandings.

<Response>: We removed ‘plant individuals’ to avoid some misunderstandings.

<Minor comment #3>: Line 135 ‘In contrast to animals and yeast’: how plants are different from them?

<Response>: The genome of Arabidopsis encodes two nucleilin-like proteins, while NUCLEOLIN is a single copy gene in animals and yeast. However, we do not describe difference of function of two NUCLEOLIN genes in this review, therefore we removed ‘In contrast to animals and yeast’ 

<Minor comment #4>: Lines 167 and 174 ‘Nucleolar enlargement’ and ‘enlarged nucleoli’: What can be deduced from this phenotype? Nucleolar/ribosomal stress symptom?

<Response>: We added the sentences as follows

Lines 397-400:

Perturbation of rRNA biogenesis generally induces structural disorders of the nucleolus, such as nucleolar enlargement in plant cells, as well as in animal cells (Ohbayashi et al., 2011; 2017a; 2017b, Nishimura et al., 2015; ref#60, 85, 86, 123). It, however, is still unknown how structural disorders of the nucleolus affects leaf development mediated by AS2-AS1.

<Minor comment #5>: Line 379: We need reference on this sentence (recently, AS2 was shown…).

<Response>: We added reference 128 (O’Malley et al., 2016; ref#129)